# Efficient Neural Architecture Transformation Search in Channel-Level for Object Detection

**Junran Peng**[1,2,3]    **Ming Sun**[2]    **Zhaoxiang Zhang**[1,3*]    **Tieniu Tan**[1,3]    **Junjie Yan**[2]

[1]University of Chinese Academy of Sciences
[2]SenseTime Group Limited
[3]Center for Research on Intelligent Perception and Computing, CASIA

## Abstract

Recently, Neural Architecture Search has achieved great success in large-scale image classification. In contrast, there have been limited works focusing on architecture search for object detection, mainly because the costly ImageNet pretraining is always required for detectors. Training from scratch, as a substitute, demands more epochs to converge and brings no computation saving. To overcome this obstacle, we introduce a practical neural architecture transformation search(NATS) algorithm for object detection in this paper. Instead of searching and constructing an entire network, NATS explores the architecture space on the base of existing network and reusing its weights. We propose a novel neural architecture search strategy in channel-level instead of path-level and devise a search space specially targeting at object detection. With the combination of these two designs, an architecture transformation scheme could be discovered to adapt a network designed for image classification to task of object detection. Since our method is gradient-based and only searches for a transformation scheme, the weights of models pretrained in ImageNet could be utilized in both searching and retraining stage, which makes the whole process very efficient. The transformed network requires no extra parameters and FLOPs, and is friendly to hardware optimization, which is practical to use in real-time application. In experiments, we demonstrate the effectiveness of NATS on networks like *ResNet* and *ResNeXt*. Our transformed networks, combined with various detection frameworks, achieve significant improvements on the COCO dataset while keeping fast.

## 1 Introduction

Convolutional neural networks have achieved significant success in recent years. With the development of better optimization and normalization methods [29, 13], many remarkable network architectures [16, 36, 38, 8, 12, 11, 35, 40, 42] have been designed for image classification based on hand-crafted heuristics. More recently, great efforts have been taken in neural architecture search(NAS) that automates the architecture design process, and noticeable results that surpass human-designed architectures have been reported in image classification [45, 46, 21, 32, 30, 24, 3].

However, there has been little works that studies NAS on backbone for object detection, mainly for two reasons: The finetuning of backbone is always necessary for detectors to converge or achieve a high performance in short time, otherwise detectors are required to be trained for much more epochs with GN [39] to reach a comparative performance according to [10]. Thus it is inefficient to directly conduct neural architecture search on object detection. Besides, the essential gap between image classification and object detection is non-negligible. The experience of NAS in image classification does not suffice for NAS in object detection, that the searching space may need to be re-defined.

---

Table 1: Comparing our method against other NAS methods. The size of training set and input size during search are given to clearly reveal the hardness of searching in different cases. Our efficient search takes only 20 1080TI GPU days on object detection even though the dataset is of large scale and input size is huge.

| Methods | Dataset | Size of Train Set | Input Size During Search | GPU-Days | Task |
|---|---|---|---|---|---|
| NASNet [46] | CIFAR-10 | 50k | $32 \times 32$ | 2000 | Cls |
| AmoebaNet [32] | CIFAR-10 | 50k | $32 \times 32$ | 3000 | Cls |
| PNASNet [21] | CIFAR-10 | 50k | $32 \times 32$ | 150 | Cls |
| EAS [2] | CIFAR-10 | 50k | $32 \times 32$ | 10 | Cls |
| DPC [5] | Cityscapes | 5k | $769 \times 769$ | 2600 | Seg |
| DARTS [24] | CIFAR-10 | 50k | $32 \times 32$ | 4 | Cls |
| ProxylessNAS [3] | ImageNet | 1.3M | $224 \times 224$ | 10 | Cls |
| Auto-Deeplab [22] | Cityscapes | 5k | $321 \times 321$ | 3 | Seg |
| NATS-det | COCO | 118k | $800 \times 1200$ | 20 | Det |

In this paper, we present effort towards practical meta-learning for object detection task to tackle these two obstacles. Instead of searching an entire network architecture [45, 46, 21, 32, 22, 5], we search for an architecture transformation strategy that adjusts the structure of existing network to fit the need of detection, and weights of pretrained model could be fully used in both searching stage and re-training stage. As demonstrated in [27], dilation of convolution layers is closely relevant to the distribution of ERFs and changing dilation does not influence the kernel size in convolution layer. Therefore a convolution layer with different dilations could reuse the pretrained weights, which makes architecture transformation on dilation-domain possible.

Additionally, unlike previous works that search for optimal paths in cell level [46, 21, 32, 30, 24, 3] or in network level [45, 31], our transformation search is conducted in channel level. To be specific, we split the forward signal generated by each path into pieces in channel domain, and treat the sub-paths as the minimum searchable units. As shown in Fig. 1, the searched path becomes a fusion of various operations with respective channels. With the combination of dilation search space and channel-level search strategy, our method, named NATS, is able to efficiently discover high-performance architecture transformation scheme for object detection.

In our experiments, NATS for detection could improve the AP of Faster-RCNN based on ResNet-50 and ResNet-101 by 2.0% and 1.8% without any extra parameters or FLOPs, and keep the inference times almost the same. The transformation is also proved to be valid for various type of detectors. On Mask-RCNN [9], Cascade-RCNN [4] and RetinaNet [20], the AP have been improved by 1.9%, 1.3% and 1.3% respectively. As shown in Table 1, the searching stage of NATS takes only 2.5 days on 8 1080TI GPUs, and retraining of searched network takes about 1 day(same as training a baseline model) with no need of extra pertraining in ImageNet [34], making the whole process efficient and practical.

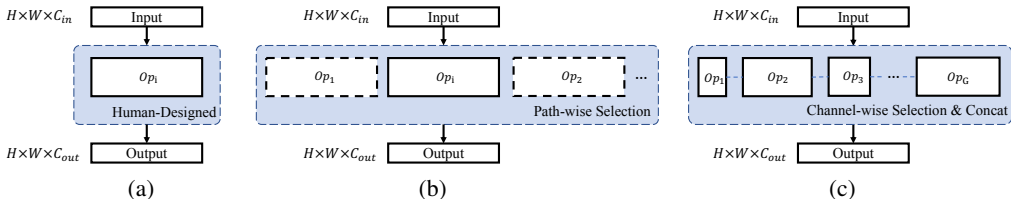

Figure 1: Different forms of operations on edge. In (a), an operation within a connection is chosen on the basis of human heuristics. For path-level search shown in (b), an operation superior to others is selected out of all operation candidates. As in (c), the path connecting input and output is decomposed into sub-paths with respective channels.

## 2 Related works

### 2.1 Object detection

Object detection is one of the most fundamental fields in computer vision for both academic research and industrial application. It aims at finding the location of each object instances and determining the categories given an image. Some fundamental works like R-CNN [7], Fast-RCNN [6], Faster R-CNN [33] and SSD [26] greatly push forward the development of this area. In general, object detectors usually consist of three parts: a backbone that takes in image as input and extract features, a neck attached to backbone that fuses or further encodes the extracted features and a head for classification and localization[2]. In the past years, great progresses have been achieved in designing each of these modules.

For backbones, there are [17, 37] designed specially for object detection manually. The deformable convolution is also proposed to enable backbone to adaptively sample input features, which is proved helpful in performance but hostile to hardware acceleration. FPN [19] is one of the representative work exploring the architecture of neck. It builds a top-down structure with lateral connections to different stages of backbone to integrate features at all scales. Many recent works [15, 14, 41] propose various multi-scale integration strategies to generate pyramidal feature representations.

In [25, 27], it is proposed that the effective receptive fields(ERFs) of backbones is essential for object detection and dilation of convolutions could effectively change the distribution of ERFs. Based on these findings, we aim to design a network architecture that holds better ERFs to handle the huge variation of object scales in detection.

### 2.2 Neural architecture search

Designing network automatically has drawn great attention recently. Several works [46, 45, 2, 1, 43] introduce reinforcement learning with RNN controller to design cell structure to form a network. In [32, 23, 28], evolution method has been used to update network structure instead of RL-based controller. These methods are sample-based that often take great amount of computational resources. In [30, 2], weights of sampled models could be reused to reduce the search cost.

Some other works tend to used gradient-based methods that search for relatively optimal child networks from predefined super-nets, which make NAS with limited computational resources possible. DARTS [24] formulates a super-network based on the continuous relaxation of the architecture representation, which allows efficient search of the architecture using gradient descent. ProxylessNAS [3] further improves the optimization strategy and imports latency loss to find more efficient architectures. In Auto-DeepLab [22], gradient-based method is also applied to search for backbone of segmentation model.

As for search space, most methods tend to search for optimal paths in cell level [46, 21, 32, 30, 24, 3] or network level [45, 31], while in this paper, we propose a novel search space in channel level. Inspired by the idea of function-preserving transformation in [2], we propose a neural architecture transformation search(NATS) algorithm to automatically find an optimal strategy to transform the structure of existing networks designed for image classification to fit task of object detection.

## 3 Methods

In this section, we first analyze a crucial factor of backbone for object detection. Then we describe our general strategy of neural architecture search in channel-domain and design a search space that enables effective architecture transformation search specialized in object detection.

### 3.1 Revisit effective receptive fields

Receptive field is one of the most basic concepts in deep CNNs. Unlike in fully connected networks that value of each neuron is associated with entire input to network, a neuron in convolutional networks depends on a certain region of the input. This property enables neurons in convolutional networks to be position-sensitive, and makes dense prediction tasks like object detection and semantic

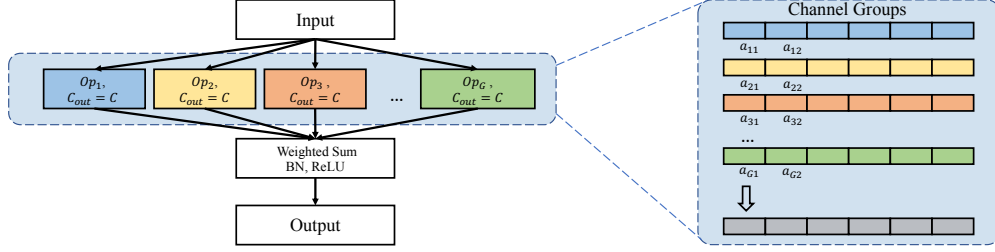

Figure 2: The structure of block during search. The output of each operation is equally divided into sub-groups in channel domain. Each sub-group of each candidate is assigned an architecture parameter to fit together as output, which makes the search space within each channel group continuous. The search between channel groups is independent.

segmentation possible. As carefully studied in [27], the distribution of impact in a receptive field is proved to be like a Gaussian and only a small central region of pixels in receptive field effectively contributes to response of neuron in output map. The region is called effective receptive field(ERF).

In tasks of image classification, the input sizes are always kept small. As in object detection, the input sizes are often much bigger and detectors are required to handle objects over a large range of scales, thus the ERFs of network designed for image classification could not suffice for this demand[3]. As mentioned in [27], changing dilations could effectively modify the ERFs distribution of convolution layers. Moreover, changing dilation does not influence the kernel size of convolution layer, which enables pretrained weights to be directly reused. Therefore in this work, we constrain our search space to dilations of convolution layers in order to grant network better ERFs for handling the huge variation of object scales.

## 3.2 Channel-level neural architecture search

A neural network is a directed acyclic graph consisting of a set of nodes connected in order. The directed edges connecting nodes are always associated with some operations that process the input signals, such as convolutional layer, max-pooling and *etc*. For most gradient-based NAS methods, an over-parameterized super-network is constructed firstly with all candidates paths included and one superior path is selected on each edge with the other candidates removed. However, signals in network often contain numerous channels during forward propagation, which means that a path is not the minimum separable structure unit in network and path-level search methods [3, 24, 22] limit the granularity of architecture search. Thus in our work, we treat a channel of signal generated by an operation of certain genotype as the minimum separable structure unit, and transform path-level NAS into channel-level NAS.

Given an input signal $x$, the output signal $y^*$ is generated based on the outputs of all $G$ candidate paths during search. Each path is associated with a certain type of operation $O^g$, and we call the categories of operation as genotypes $\mathcal{G}$ with $g \in \mathcal{G}$. While in DARTS and Auto-Deeplab, each entire path is assigned an architecture parameter $\alpha^g$ and $y^*$ is weighted sum of input signals where the weights are calculated by applying softmax to $\alpha^g$:

$$y^g = O^g(x), \quad y^* = \sum_{g \in \mathcal{G}} \frac{exp(\alpha^g)}{\sum_{g' \in \mathcal{G}} exp(\alpha^{g'})} y^g \tag{1}$$

After obtaining the continuous super-architecture with $\alpha$, every edge with mixed operation of all genotypes is replaced with the most likely operation by taking the argmax of $\alpha^g$. Thus only one genotype is selected to handle input signals on each edge in the outcome architecture.

To apply a more fine-grained architecture search, we equally divide $y^g$ into $N$ groups in channel domain for each genotype as follows:

$$y^g \Rightarrow \{y_1^g, y_2^g, ..., y_i^g, ..., y_N^g\}, \quad with \ C_{out} = \sum_i^N C_i^g, \tag{2}$$

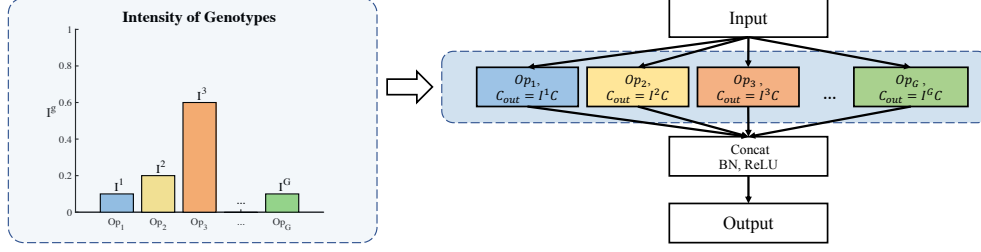

Figure 3: Decoding discrete architecture based on intensity of genotypes.

where $i$ denotes the index of channel group and $C_{out}$ denotes the total output channels. As illustrated in Fig. 2, instead of assigning path-wise architecture parameters we assign each channel group an architecture parameter $\alpha_i^g$ where $1 \leq i \leq N$. We use the continuous relaxation among genotypes in each channel group and the output of group $i$ is obtained as:

$$y_i^* = \sum_{g \in \mathcal{G}} \frac{exp(a_i^g)}{\sum_{g' \in \mathcal{G}} exp(a_i^{g'})} y_i^g \tag{3}$$

In this way, the super-net is constructed in which nodes are connected with sub-paths in channel domain and architecture parameters $\alpha_i^g$ are learnt for each genotype in each channel group. The training set is divided into two splits, and the optimization alternates between updating network parameters in the first split and updating architecture parameters $\alpha_i^g$ in the other split.

### 3.3 Decoding discrete architectures with channel decomposition

Unlike [24], [22] and [3]that select path with the maximum probability and prune redundant paths, the discrete architecture decoding in our method is conducted based on the distribution of $\alpha_i^g$. We first keep the index of genotype with the maximum probability in each channel group as

$$ind_i = \arg\max_g \alpha_i^g, \tag{4}$$

and calculate the intensity of each genotype throughout all channel groups as:

$$I^g = \frac{\sum_i^N 1(ind_i = g)}{N} \tag{5}$$

As illustrated in Fig. 3, we retain all the paths that have a positive $I^g$ but reset output channels according to $I^g$ as $\tilde{C}_{out}^g = C_{out}I^g$. The output feature maps of different genotypes are concatenated together to form a final output $y$ as follows:

$$\{y^1, y^2, ..., y^g, ..., y^G\} \Rightarrow y \tag{6}$$

### 3.4 Architecture transformation search for object detection

Taking bottleneck structure in ResNet as example in our paper, the transformation search is applied on the $3 \times 3$ convolution layer in the middle. Dilations in both orientation of the convolution $\{d_h, d_w\}$ is set as our search space. Since changing dilations does not modify the kernel size or the shape of weights, we could directly transfer weights of pretrained model to our networks in both searching stage and retraining stage Combining the channel-domain searching strategy with the dilation search space makes our neural architecture transformation search possible. The whole process is gradient-based and extra pretraining is of no need, which makes our method very efficient.

During the training of super-network, backbone is initialized with the weights pretrained on ImageNet. For each $3 \times 3$ convolution layer in stage-3,4,5, weights are copied to all of its dilated replicas. Weight initialization for searched model is different in re-training stage. Since the original $3 \times 3$ convolution layer has been decomposed into sub-convs with various dilations and output channels, the pretrained weight $W$ with shape $C_{out} \times C_{in} \times K \times K$ is also decomposed into $G$ groups with shape $\{C_{out}^g \times C_{in} \times K \times K\}_{g=1}^G$ in order to fit the weights shape of sub-convs.

Table 2: Performance on *minival* with fixed number of channel groups for NATS. When number of groups is 1, the architecture transformation search is on path-level.

| Num of Groups | AP | $AP_{50}$ | $AP_{75}$ | $AP_S$ | $AP_M$ | $AP_L$ |
|---|---|---|---|---|---|---|
| baseline | 36.4 | 58.9 | 38.9 | 21.4 | 39.8 | 47.2 |
| 1 | 36.9 | 58.9 | 39.1 | 21.3 | 40.1 | 47.5 |
| 2 | 37.2 | 59.6 | 39.8 | 21.6 | 40.8 | 48.9 |
| 4 | 37.9 | 60.2 | 40.9 | 22.2 | 40.9 | 49.9 |
| 8 | 37.8 | 60.4 | 40.4 | 21.4 | 41.3 | 50.0 |
| 16 | 38.4 | 61.0 | 41.2 | 22.5 | 41.8 | 50.4 |
| 32 | 38.2 | 60.6 | 41.0 | 22.3 | 41.7 | 50.1 |

Table 3: Performance on *minival* with fixed number of channels per group for NATS.

| Channels Per Group | AP | $AP_{50}$ | $AP_{75}$ | $AP_S$ | $AP_M$ | $AP_L$ |
|---|---|---|---|---|---|---|
| baseline | 36.4 | 58.6 | 38.6 | 21.0 | 39.8 | 47.2 |
| 1 | 38.0 | 60.5 | 40.5 | 22.5 | 41.4 | 50.3 |
| 8 | 38.1 | 60.7 | 40.7 | 22.3 | 41.6 | 50.2 |
| 16 | 38.2 | 60.7 | 40.9 | 22.4 | 41.6 | 50.5 |
| 32 | 38.3 | 60.9 | 41.3 | 22.3 | 41.9 | 50.4 |
| 64 | 37.8 | 60.5 | 40.4 | 21.7 | 40.9 | 50.3 |

# 4 Experiments and results

## 4.1 COCO dataset

We use the MS-COCO [18] for experiment in this paper. It contains 83K training images in *train*2014 and 40K validation images in *val*2014. In its 2017 version, it has 118K images in *train*2017 set and 5K images in *val*2017(a.k.a *minival*). The dataset is widely believed challenging in particular due to huge variation of object scales and large number of objects per image. We consider AP@IoU as evaluation metric which averages mAP across IoU threshold ranging from 0.50 to 0.95 with an interval of 0.05. During searching stage, we use *train*2014 for training model parameters and use 35K images from *val*2014 that are not in *minival* for calibrating architecture parameters. During retraining stage, our searched model is trained with *train*2017 and evaluated with *minival* as convention.

## 4.2 Implementation details

In our method we firstly search for an appropriate structure transformation scheme on COCO2014 dataset, then we train our searched model on COCO2017 dataset as mentioned above. We experiment on the Faster-RCNN baselines with FPN [19], and adopt models pretrained in ImageNet [34] for weight initialization in both searching and training stages.

**Searching details.** We conduct architecture transformation search for 25 epochs in total. To make the super-network converge better, architecture parameters are designed not to be updated in the first 10 epochs. The batch size is 1 image per GPU due to GPU memory constraint. We use SGD optimizer with momentum 0.9 and weight decay 0.0001 for training model weights. Cosine annealing learning rate that decays from 0.00125 to 0.00005 is applied as lr-scheduler. When training architecture parameters $\alpha$, we use Adam optimizer with learning rate 0.01 and weight decay 0.00001.

**Training details.** After the architecture searching is finished, we decode discrete architecture as mentioned in 3.3. We use SGD optimizer with 0.9 momentum and 0.0001 weight decay. For fair comparison, all our model is trained for 13 epochs, known as $1\times$ schedule. The initial learning rate is set 0.00125 per image and is divided by 10 at 8 and 11 epochs. Warming up and Synchronized BatchNorm mechanism are applied in both baselines and our searched models for multi-GPU training. It takes approximately 2.5 days to finish the search for 8 1080TI GPUs.

## 4.3 Object detection results

In our paper, ResNet[8] and ResNeXt[40] are selected as backbone in all experiment settings. Following the regime mentioned in DCNv2 [44], we apply architecture transformation search only

Table 4: Performance of NATS on ResNet101 and ResNeXt101. NATS is conducted with fixed number of channel per group as $C = 32$ in this ablation study.

| Backbone | AP | $AP_{50}$ | $AP_{75}$ | $AP_S$ | $AP_M$ | $AP_L$ |
|---|---|---|---|---|---|---|
| R101 | 38.6 | 60.7 | 41.7 | 22.8 | 42.8 | 49.6 |
| R101-NATS | **40.4** | **62.6** | **44.0** | **23.2** | **44.1** | **53.3** |
| X101-32×4d | 40.5 | 63.1 | 44.1 | 24.2 | 45.1 | 52.9 |
| X101-32×4d-NATS | **41.6** | **64.3** | **45.2** | **24.9** | **45.5** | **54.8** |

Table 5: Comparison between different sets of genotypes on COCO *minival*.

| GENOs | AP | $AP_{50}$ | $AP_{75}$ | $AP_S$ | $AP_M$ | $AP_L$ |
|---|---|---|---|---|---|---|
| baseline | 36.4 | 58.9 | 38.9 | 21.4 | 39.8 | 47.2 |
| NATS-A | 37.7 | 59.9 | 40.5 | 22.0 | 40.8 | 49.8 |
| NATS-B | 38.0 | 60.5 | 40.7 | 21.8 | 41.4 | **50.5** |
| NATS-C | **38.4** | **61.0** | **41.2** | **22.5** | **41.8** | 50.4 |

on blocks of stage-3,4,5 in backbone. For stage-3 and stage-4, the dilation candidates are {1, 2, 3, (1,3), (3,1)}. The dilation candidates of stage-5 are {1, 2, 3, 4, 5, (1,3), (3,1), (1,5), (5,1)}. No extra parameters or FLOPs is imported in our transformed architectures.

**Group division.** We evaluate different ways of dividing output channels into groups. With a given fixed group number($G \in \{1, 2, 4, 8, 16, 32\}$, NATS is applied on ResNet-50. In Table 2, we find that more groups could achieve better performance. With a fixed group number of 16, the transformed architecture achieves an AP of 38.4%(2.0% higher than the baseline). Note that $G = 1$ is a special case which is the path-level searching strategy similar to DARTS [24] and ProxylessNAS [3], and the improvement is limited(only 0.5% over baseline).

We also fixed the number of channels($C \in \{1, 8, 16, 32, 64\}$ per searching group. Since different blocks have different channel numbers, the group number can change across layers in this setting. The results are shown in Table 3. With a fixed channel number per searching group, our transformed ResNet-50 achieves a *minival* AP of 38.3% which is 1.9% higher than baselines.

From both setting we find that searching with a more fine-grained grouping is better in general. We infer that it enables blocks to have more combinations of operations with different dilations, which brings more flexible ERFs. We also find that the improvement of AP increases as scale of objects grows. In model searched with $G = 16$, the improvements of $AP_S$, $AP_M$ and $AP_L$ are 1.5%, 2.0% and 3.2% respectively.

**Deeper models.** It is known that deeper networks have larger ERFs with stronger intensity and may dilute the effects of many approaches, thus we study the impact of architecture transformation on deeper networks. We have compared transformed backbones with baselines on ResNet-101 and ResNeXt-101. As shown in table 4, architecture transformation yields 1.8% AP improvement on ResNet-101, from 38.6 to 40.4. While in ResNeXt-101, we use the $32 \times 4d$ configuration and the channels per group is set 32 to be consistent with backbone. Architecture transformation yields 1.1 improvement from 40.5 to 41.6. Comparing ResNet-101 with shallower network like ResNet-50, the improvement of $AP_S$ is relatively small(0.4% *v.s.* 0.7%), but improvement of $AP_L$ is even greater(3.7% *v.s.* 3.2%) even through it is deeper. ResNeXt-101 acts also in the similar way.

**Influence of genotypes.** In this section we explore the influence of genotypes included during arch-transformation search. We include different set of dilation candidates as genotypes in this ablation study. We first investigate the necessity of dense dilation candidates. For stage-3,4,5 we set the dilation candidates {1, 3}, {1, 3}, {1, 3, 5} respectively as setting A, and set the dilation candidates {1, 2, 3}, {1, 2, 3}, {1, 2, 3, 4, 5} as setting B. To explore the influence of ratio aspects we add (1, 3),(3, 1) for stage-3,4 and (1, 3), (3, 1), (1, 5), (5, 1) for stage-5 as candidates in setting C. As shown in Table 5, NATS-B is higher than PATS-A by 0.3%, which implies that denser dilation candidates is slightly better. NATS-C is 0.4% better than PATS-B, demonstrating that dilations with aspect ratios are beneficial for object detection.

Table 6: Comparison of performance of NATS on different type of detectors.

| Method | Backbone | AP | AP$_{50}$ | AP$_{75}$ | AP$_S$ | AP$_M$ | $AP_L$ |
|---|---|---|---|---|---|---|---|
| Faster-RCNN | R50 | 36.4 | 58.9 | 38.9 | 21.4 | 39.8 | 47.2 |
| Faster-RCNN | R50-NATS | **38.4** | **61.0** | **40.8** | **22.1** | **41.5** | **50.5** |
| Mask-RCNN | R50 | 37.5 | 59.6 | 40.5 | 22.0 | 41.0 | 48.4 |
| Mask-RCNN | R50-NATS | **39.3** | **61.3** | **42.6** | **23.0** | **42.5** | **51.7** |
| Cascade-RCNN | R50 | 40.7 | 59.4 | 44.2 | 22.9 | 43.9 | 54.2 |
| Cascade-RCNN | R50-NATS | **42.0** | **61.4** | **45.5** | **24.2** | **45.3** | **55.9** |
| RetinaNet | R50 | 36.0 | 56.1 | 38.6 | 20.4 | 40.0 | 48.2 |
| RetinaNet | R50-NATS | **37.3** | **57.8** | **39.5** | **20.7** | **40.8** | **49.6** |

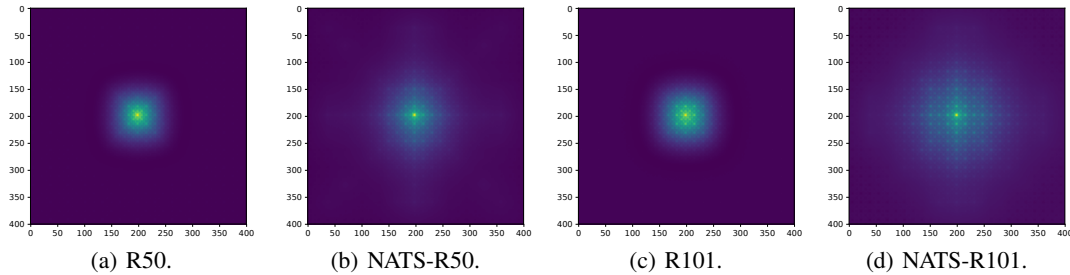

(a) R50.     (b) NATS-R50.     (c) R101.     (d) NATS-R101.

Figure 4: Visualization of ERFs in transformed architectures and vanilla architectures.

**Various detectors.** To validate the generalization ability of our method, we also combine the transformed networks with different type of detectors. Several well-known and remarkable frameworks like Mask-RCNN [9], Cascade-RCNN [4] and RetinaNet [20] are selected in this ablation study. ResNet-50 and the transformed ResNet-50(G=16) are selected as backbones and all the models are trained with $1\times$ lr-schedule. As demonstrated in Table 6, performances of all detectors in chart are improved prominently($1.8\%$ in Mask-RCNN, $1.3\%$ in Cascade-RCNN and $1.3\%$ in RetinaNet). This shows the strong generalization capability of networks searched through our transformation method.

## 4.4 Visualization of ERFs

Following the regime mentioned in [27], we visualize the receptive field of neuron on map of the last convolution layer. The input values are set 1 for whole image and only the neuron in the center of output map propagates backward. To focus on only the intensity of connections, ReLUs are abandoned during visualization. As shown in the Fig. 4, the size of ERFs in our transformed network are larger than ERFs of the vanilla structures. While the intensities in the center region are kept strong, the intensities of outer region becomes weaker as the region becomes bigger. It indicates that this type of ERFs could better fit the task of object detection.

## 5 Conclusion

In this paper, we present NATS that can efficiently learn an neural architecture transformation strategy to adapt existing networks to new tasks. We propose a novel architecture search scheme in channel domain and design a search space of dilations targeting at object detection, which makes the neural architecture transformation search possible. Finetuning from pretrained models is feasible in both searching and re-training stages, making the whole process very efficient. Experiments on the COCO dataset have demonstrated that NATS could effectively improve the capability of networks to handle huge variation of object scales and robustly yield improvements on various type of detectors. In the future, we would like to investigate architecture transformation search on depth and width of each stage for object detection task.

# 6 Acknowledgements

This work was supported in part by the National Key R&D Program of China(No.2018YFB-1402605), the Beijing Municipal Natural Science Foundation (No.Z181100008918010), the National Natural Science Foundation of China(No.61836014, No.61761146004, No.61773375, No.61602481) and CAS-AIR.

## Footnotes

[2]For one-stage detectors, the form of head is fully convolution.

[3]Given a conventional input size of $800 \times 1200$, size of objects varies from 32 to 800 pixels in COCO, while the size of ERFs in ResNet50 is approximately 100 pixels as shown in 4(a).

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
