[Supplementary Material]

# Appendix

# 1 Searched transformation strategies

We display the channels of each block in the transformed network architecture.

Table 1: Setting of searched ResNet-50(G=16) transformation.

| Layer Name | Groups | (1,1) | (2,2) | (3,3) | (4,4) | (5,5) | (1,3) | (3,1) | (1,5) | (5,1) |
|---|---|---|---|---|---|---|---|---|---|---|
| 3-1 | 16 | 32 | 32 | 0 | - | - | 56 | 8 | - | - |
| 3-2 | 16 | 32 | 32 | 32 | - | - | 8 | 24 | - | - |
| 3-3 | 16 | 88 | 24 | 16 | - | - | 0 | 0 | - | - |
| 3-4 | 16 | 64 | 16 | 16 | - | - | 24 | 8 | - | - |
| 4-1 | 16 | 64 | 48 | 48 | - | - | 32 | 64 | - | - |
| 4-2 | 16 | 64 | 16 | 112 | - | - | 16 | 48 | - | - |
| 4-3 | 16 | 48 | 16 | 64 | - | - | 96 | 32 | - | - |
| 4-4 | 16 | 80 | 0 | 16 | - | - | 80 | 80 | - | - |
| 4-5 | 16 | 64 | 32 | 48 | - | - | 32 | 80 | - | - |
| 4-6 | 16 | 64 | 112 | 16 | - | - | 32 | 32 | - | - |
| 5-1 | 16 | 96 | 96 | 64 | 64 | 0 | 0 | 0 | 32 | 160 |
| 5-2 | 16 | 64 | 128 | 32 | 0 | 96 | 0 | 64 | 96 | 32 |
| 5-3 | 16 | 128 | 32 | 32 | 96 | 32 | 96 | 64 | 32 | 0 |

# 2 Other evidence

The training and evaluation log of different type of detectors are provided, which are *frcnn-nats-r50.log*, *mask-rcnn-nats-r50.log*, *cascade-rcnn-nats-r50.log* and *retina-nats-r50.log*. Training setting, loss of each iteration, training time and inference time are included in log.

The log *frcnn-baseline-eval.log* is provided to prove that our transformation does not influence inference time at all.