[Reviews · NeurIPS 2019]

Reviewer 1



The paper reads very well and manages to present both the challenges of NAS and the proposed idea in a very understandable form (although English grammar and spelling could be improved). The paper's main idea is to constrain the search space of NAS to the dilation factor of convolutions, such that the effective receptive field of units in the network can be varied, while keeping the network weights fixed (or at least allowing the weights to be re-used and smoothly varied during the optimization). This idea is very attractive from a computational point of view, since it allows the notoriously expensive NAS process to achieve faster progress by avoiding the need for ImageNet pre-training after every architecture change. On the flip side, the proposed NATS method only explores part of the potential search space of neural architecture variations. So, the longer-term effect will depend on how restrictive this choice of search space is. I.e., do we lose on potential object detection performance by only exploring the dilation factor of convolutions and would other architecture variations achieve larger improvements? This question is very hard to answer, since (if the paper is correct) no alternative NAS methods are available that could deal with the overhead of ImageNet pre-training. Apart from this issue, the paper presents a very convincing experimental validation. It demonstrates consistent performance improvements across different backbones (ResNet50, ResNet101, ResNeXt101) and in combination with different detector heads (Faster-RCNN, Mask-RCNN, Cascade-RCNN, RetinaNet). An ablation study explores the influence of different numbers of channel groups (Tab.2), channels per group (Tab.3), and different dilation densities and aspect ratios (Tab.5). Questions: - What is the computational cost of performing the optimizations presented in Tab. 4 and 5? Does each NATS row of the table correspond to 20 GPU days worth of computation (L204), or were there larger variations for the different backbone architectures? - As stated in L193, the architecture transformation search is performed for 25 epochs in total, while the architecture parameters are designed not to be updated for the first 10 epochs to achieve better convergence. I was surprised to see such a relatively low number of epochs here. Although one epoch corresponds to a run over the full COCO training set, I would have expected more epochs to be necessary for optimization. Is this low number a restriction in practice? - What was actually the outcome of the optimization? I.e., in the optimibzed network architectures, what were the modifications that ended up being selected? Is it possible to derive some general insights into what are good architectural features for object detection? In particular, was there an observable trend in how dilation factors varied across the different layers of the network (and did this change for deeper architectures)? Update: The rebuttal cleared up my remaining questions. I think this is good work that should be published. In the long run, the point about the necessity of pretraining R5 brought up would indeed be interesting to explore. However, even as it is, the paper already provides a very useful point of reference that future papers on NAS for objection can refer to and compete against. I keep my vote to Accept.

Reviewer 2



- The proposed search space is new and the model transformation method for practical NAS for object detection task is interesting. However, the search pipeline is largely based on previous works on gradient-based NAS, which makes the novelty of the proposed method limited. - The paper is well written and easy to follow. The authors clearly outline the problem that they are trying to solve, and the experiments look reasonable. - Experiments on various detectors clearly show the effectiveness of the proposed search method and search space. - The efficiency of the discovered method is not discussed although it is mentioned in line 63: "and keep the inference times almost the same". The log file provided in supplementary material is also not clear enough. Since adding extra paths and dilated convolution may introduce considerable computational overhead, it is important to provide a detailed runtime report on inference speed to further validate the efficiency. Update: The rebuttal addressed my concern on the efficiency of the discovered models. Although I still think the technical contribution of this paper is limited, I agree with the other reviewers that the proposed practical NAS method for objection detection can be a new and promising direction in future research. I think this is a good paper overall. Therefore, I would like to revise my score to 6.

Reviewer 3



- Originality: This paper addresses an important problem - NAS for object detection. In particular, it identifies the need to use pre-training as. a key obstacle towards NAS for detection. This is a new problem that has not been considered previously in NAS and object detection research. UPDATE: - Quality: This paper has a good structure and a thorough literature survey. However, in some cases it seems to have omitted some related works, especially in object detection. More specifically. This work claims object detection must use “pre-training”. While this is mostly true, prior works on getting rid of pre-training should be acknowledged. As an example: [1]: Zhu et al. “ScratchDet: Training Single-Shot Object Detectors from Scratch”, CVPR 19 oral. In the discussion, the paper also omits that Auto-Deeplab [22] essentially argues that running NAS from scratch on segmentation can essentially match the performance of the use of a pre-trained model. While this may not be the case for object detection (although it is hard to believe so, since segmentation is similar to object detection in many aspects), the current presentation which seems to be asserting “pre-training” as “absolutely necessary” is trivializing the more subtle topic. - Clarity: The paper has good use of language in general. The motivation, literature survey and experiments section are clear and informative. However, there are major issues in the presentation of methodology: Line 148-149: the meaning of “i” is not clearly defined, although it is likely to be channel index. Equation 2: C_i^g and C_{out} are not defined. Equation 4: “I^g” is not well defined. Also, ind_i is defined using vague language rather than precise mathematical statement which is essential for such a critical part of the presentation. Equation 5: It is not clear how “I^g” affects “y^g”...This may again due to the lack of precise definition of “I^g”. In general, with the help of Figure 2, 3 and the simplicity of the method in general, the core idea, namely the paper proposes to assign different dilation rates at a per-channel level is clear. However, the presentation is not clear enough for a reader to re-implement the method. - Significance: - The paper addresses an important question. The improvement is convincing suggesting that it is a practical method for NAS in object detection. There are a few advantages in this method that makes it likely to inspire future methods: (1) It is quite efficient, as shown in comparison in Table 1, although efficiency is derived from the continuous relaxation in DARTS and thus not an original contribution. (2) It does not introduce additional FLOPs, since it focuses on dilation operators. (3) It can use pre-trained models, which again stems from the use of dilation. - There are important claims that are not justified, which is a limiting factor in its significance. Note that the paper propose “channel-wise” search as an important contribution. However, no direct comparison with “path-level” search is provided. This shouldn’t be impossible, as the proposed method is essentially adding degrees of freedom in the architecture parameters so that they are channel wise. The lack of ablations in this regard limits the significance of this work. UPDATE: The rebuttal fails to convince me that the discussions on pre-training is fair and complete. This is important but not critical to me. The clarification of methodology and comparisons with path-level search are on the other hand critical, and I see a much better case there. I revise my score up to 6.

[Author Response · NeurIPS 2019]

# Responses to Reviewer #2

**Q1:** *Please provide computational cost of searching networks in Tab4&5.*

**A1:** In Table 4, NATS on R101 and X101-32×4d take 27 and 36 GPU-days respectively. In Table 5, NATS-A, NATS-B and NATS-C on R50 take 15, 17 and 20 GPU-days respectively. Generally, larger base-networks or more genotypes would require more search time.

**Q2:** *Please explain about fixation of arch-parameters for the first 10 epochs.*

**A2:** We find in experiments that searching without fixing architecture for awhile may lead the hyper-net to converge to sub-optimal state, because random paths may take over and prevent other paths to learn in early stage. Fixing arch-parameters for several epochs greatly alleviate the problem. 10 epochs of arch-fixation is an appropriate option.

**Q3:** *Please provide the outcome of optimization and explain.*

**A3:** We list one of the optimized network architecture in the supplementary material. As shown in Table1 in the supplementary material, each convolution tends to have various dilation types and large dilations seem to be helpful. We infer that it is because the detector has to deal with objects of large scale variation.

# Responses to Reviewer #4

**Q1:** *Please provide results about efficiency of the discovered models.*

**A1:** As shown in Table A, B-G16 only takes 6 extra ms but yields 2AP improvement compared to baseline.

**Q2:** *The novelty of the proposed method is limited.*

**A2:** To our best knowledge, we are the first to explore this group-wise search space. 1) **Efficiency.** Our search space is compatible with searching and re-training with pre-trained models, which improves the search efficiency to a large extent. 2) **Effectiveness.** Our search space is proved to effectively improve the performance in task of object detection. Besides, a very recent work[2] of Google Brain has also verified the effectiveness of this search space in image classification, while we are studying it **before them** in this more complicated object detection task.

Table A: Inference time of R50 backbones. **B-Gn** means the backbone is searched with group number **n**.

| Backbone | B(Baseline) | B-G1 | B-G2 | B-G4 | B-G8 | B-G16 | B-G32 |
|---|---|---|---|---|---|---|---|
| Inference Time(ms) | 42 | 44 | 45 | 47 | 48 | 48 | 48 |

# Responses to Reviewer #5

**Q1:** *Please improve the presentation in methodology.*

**A1:** We are sorry for our unclear presentation. 1) L148-149: the meaning of $i$ is index of the $i$-th channel group; 2) Equation 2: $C_{out}$ means the output channel of a path and $C_i^g$ means the output channel of the $g$-th genotype in its $i$-th channel group; 3) We totally agree with your precious suggestion, and $ind_i$ is now defined as $ind_i = \arg\max_g \alpha_i^g$; 4) With the definition of $ind_i$, intensity of each genotype $I^g$ is defined as equation 4. And the output channel of $y^g$ is obtained as $C^g = C_{out}I^g$. We would further improve our presentation to make our paper better. Thank you.

**Q2:** *About the influence of pre-training in object detection.*

**A2:**

**1) Pre-training in detection.** We does not claim that *object detection must use pre-training* and explain the influence of pre-training in object detection in L38-40. As explored in [1], training from scratch in object detection is **feasible but requires multi-fold extra training time** to reach a comparable performance. We show our results to support this conclusion in Table B. In [3] you mentioned(we would add it in reference), detectors are also trained with multi-fold training time(84.6 vs. 29.7 hours) which is in accordance with our point.

**2) Reduce the search time.** Learning from scratch in NAS would require even more time while our search space is compatible with searching based on pre-training which greatly accelerates the search of object detector.

Table B: Faster-RCNN with FPN of different training schedules. **n**×: $n \times 13$ training epochs. **ft**: finetuning.

| Backbone | R50-1x-scratch | R50-1x-ft | R50-2x-scratch | R50-2x-ft | R50-6x-scratch | R50-6x-ft |
|---|---|---|---|---|---|---|
| COCO-AP | 33.2 | 36.4 | 34.5 | 37.8 | 37.9 | 38.0 |

**Q3:** *Please compare channel-level search with path-level search.*

**A3:** The result of path-level search in listed in the second row of Table 2 in our paper. Please refer to L214-L216. It is shown that path-level search in this setting is less effective. We infer that a single dilation type for each layer might be insufficient to handle the huge scale variation of objects compared to mixed dilation types.

# References

[1] K. He, R. Girshick, and P. Dollár. Rethinking imagenet pre-training. *arXiv preprint arXiv:1811.08883*, 2018.

[2] M. Tan and Q. V. Le. Mixnet: Mixed depthwise convolutional kernels. *arXiv preprint arXiv:1907.09595*, 2019.

[3] R. Zhu, S. Zhang, X. Wang, L. Wen, H. Shi, L. Bo, and T. Mei. Scratchdet: Training single-shot object detectors from scratch. In *CVPR*, 2019.


[Meta-Review · NeurIPS 2019]

This paper proposes a neural architecture search method specifically for object detection tasks. Although the review scores were initially borderline, the feedback and the subsequent discussion swayed the reviewers into a jointly and consistently positive opinion of the paper. Although the concerns of R5 remain, even this reviewer agrees that they are not sufficient to criticise this work as a whole. I thus recommend to accept this paper.